# Harnessing Quantum Capacitance in 2D Material/Molecular Layer Junctions for Novel Electronic Device Functionality

**DOI:** 10.3390/nano14110972

**Published:** 2024-06-03

**Authors:** Bhartendu Papnai, Ding-Rui Chen, Rapti Ghosh, Zhi-Long Yen, Yu-Xiang Chen, Khalil Ur Rehman, Hsin-Yi Tiffany Chen, Ya-Ping Hsieh, Mario Hofmann

**Affiliations:** 1Department of Engineering and System Science, National Tsing Hua University, Hsinchu 300044, Taiwan; bharatpapnai@gmail.com (B.P.); hsinyi.tiffany.chen@gapp.nthu.edu (H.-Y.T.C.); 2Nanoscience and Technology Program, Taiwan International Graduate Program, Academia Sinica, Taipei 10617, Taiwan; 3Department of Physics, National Taiwan University, Taipei 10617, Taiwan; 4Institute of Atomic and Molecular Sciences, Academia Sinica, Taipei 10617, Taiwan; dingruichen54@gmail.com (D.-R.C.); d12551007@ntu.edu.tw (Z.-L.Y.); d11551008@ntu.edu.tw (Y.-X.C.); khalilmrwt@gmail.com (K.U.R.); yphsieh@gate.sinica.edu.tw (Y.-P.H.); 5Department of Electrical Engineering and Computer Sciences, Massachusetts Institute of Technology, Cambridge, MA 02139, USA; 6Pritzker School of Molecular Engineering, University of Chicago, Chicago, IL 60637, USA; raptigh28@gmail.com; 7Chemical Sciences and Engineering Division, Physical Sciences and Engineering Directorate, Argonne National Laboratory, Lemont, IL 60439, USA; 8International Graduate Program of Molecular Science and Technology, National Taiwan University, Taipei 10617, Taiwan; 9Molecular Science and Technology Program, Taiwan International Graduate Program, Academia Sinica, Taipei 10617, Taiwan; 10College of Semiconductor Research, National Tsing Hua University, Hsinchu 30013, Taiwan; 11Department of Materials Science and Engineering, National Tsing Hua University, Hsinchu 30013, Taiwan

**Keywords:** 2D materials, negative differential resistance, Langmuir–Blodgett, molecular layer, quantum capacitance

## Abstract

Two-dimensional (2D) materials promise advances in electronic devices beyond Moore’s scaling law through extended functionality, such as non-monotonic dependence of device parameters on input parameters. However, the robustness and performance of effects like negative differential resistance (NDR) and anti-ambipolar behavior have been limited in scale and robustness by relying on atomic defects and complex heterojunctions. In this paper, we introduce a novel device concept that utilizes the quantum capacitance of junctions between 2D materials and molecular layers. We realized a variable capacitance 2D molecular junction (vc2Dmj) diode through the scalable integration of graphene and single layers of stearic acid. The vc2Dmj exhibits NDR with a substantial peak-to-valley ratio even at room temperature and an active negative resistance region. The origin of this unique behavior was identified through thermoelectric measurements and ab initio calculations to be a hybridization effect between graphene and the molecular layer. The enhancement of device parameters through morphology optimization highlights the potential of our approach toward new functionalities that advance the landscape of future electronics.

## 1. Introduction

Two-dimensional materials have received significant attention in future electronics due to their high carrier mobility and crystalline quality at atomic dimensions. Their wide range of compositions and unique properties make them particularly well-suited for electronic functionality in the “more-than-Moore” era [1].

To realize functionality beyond conventional electronics, non-monotonic effects have received significant attention. Compared to conventional electronics, non-monotonic effects introduce complexities that enable unprecedented functionalities and capabilities. Negative Differential Resistance (NDR), for instance, exhibits regions where the current decreases with increasing voltage and offers the potential for ultrafast switching speeds and reduced power consumption, which are essential in a modern electronic system [2]. Memristive devices emulate synaptic plasticity and exhibit non-monotonic hysteresis that enable next-generation memory and neuromorphic computing systems [3]. Non-monotonic effects have been investigated in 2D materials recently [4]. Zhao et al. characterized the NDR arising from resonant tunneling through defects in hBN. Roshini et al. produced bi-anti-ambipolar transconductance by adjusting the contribution of basal plane and edge injection in strained InSe/SnSe2 heterojunctions [5].

Unfortunately, these approaches require careful control of 2D materials’ properties, the introduction of atomic defects, or high-quality heterojunctions, thus limiting their robustness. Moreover, the limited scalability of the device fabrication restricts the commercial impact of such devices.

We here introduce a novel device concept that we term variable capacitance 2D molecular junction (vc2Dmj). Through a combination of scalably producible graphene and a molecular dielectric, a heterojunction was realized that exhibits pronounced NDR and even active resistance control (ANR). The device operation relies on the variable and low quantum capacitance of graphene and shows high robustness at room-temperature operation and in the presence of device geometry variation. Our analysis found a peak-to-valley ratio of 2.5, which demonstrates the potential of our approach. Finally, the thermoelectric characterization revealed a high thermoelectric performance that opens up new routes toward powering future electronics.

## 2. Materials and Methods

Graphene was produced by chemical vapor deposition, following previous reports [6]. Briefly, copper foil was placed into a clamshell furnace (1” diameter) before flowing argon and hydrogen, while maintaining a low pressure (100 mTorr). Methane (CH_4_) was utilized as a carbon-containing precursor that was introduced at elevated temperatures (1000 °C). After 6 h of growth, the sample was cooled naturally to room temperature under Ar and H_2_ flow.

The stearic acid was produced by dissolving stearic acid (1 mg) in hexane (1 mL) and then compressed using the Langmuir–Blodgett through from KSV NIMA Instruments (Biolin Scientific, Vastra Frolunda, Sweden). By spreading the hexane solution of stearic acid of 80 µL by using a microsyringe (Hamilton Company Inc., Reno, NV, USA) on DI water with a resistivity ≥18.2 MΩ cm, the monolayers were formed. The isothermal compression rate was maintained at 10 mm/min, and once the desired surface tension was achieved, the monolayer was allowed to equilibrate for about 5 min. Then, we employed the LB deposition technique to transfer a single layer or multilayer of stearic acid on the substrate (graphene over SiO_2_ wafer) through withdrawing them vertically out of the subphase at the rate of 5 mm/min. The quality, structure, and behavior studies of the monolayer were performed with a KSV NIMA Micro-BAM Brewster Angle Microscope (Biolin Scientific, Biolin Scientific, Vastra Frolunda, Sweden). A Wilhelmy plate was used to measure the surface pressure.

To measure the electric properties of the device, a probe station with four manual manipulators (Keithlink Technology Co., Ltd., Taipei, Taiwan) was used. Electrical contact to the device under test was made through four tungsten probes in micromanipulators that were connected to a semiconductor analyzer (HP-4156B by Hewlett-Packard Company, Fort Collins, CO, USA). Temperature-dependent IV characteristics were conducted in a Janis probe station under vacuum (<1E-6Torr), while other measurements were conducted in an ambient environmental and light conditions.

Morphology measurements were conducted by Atomic Force microscopy (VEECO by Veeco Taiwan Co., Hsinchu, Taiwan) and optical microscopy (Olympus BX53 by Evident Corp. and Olympus Scientific Solutions Americas Corp., Tokyo, Japan). Raman spectra were collected by a NANOBASE XPER RF Raman system (NBOS-220012) by Nanobase Co., Seoul, Republic of Korea with a 532 nm excitation laser.

Ab initio simulations were conducted using QuantumATK (Synopsys Taiwan Co., Ltd., Hsinchu, Taiwan). We utilized a computational LCAO basis set to calculate the density of state for both stearic acid and the stearic acid/graphene junction.

## 3. Results

The proposed vc2Dmj exploits the quantum capacitance of a graphene channel that is electrostatically floating between a source and drain contact (Figure 1a). For this purpose, graphene was separated from Au electrodes through a molecular film of stearic acid. Stearic acid consists of a linear chain with 18 carbon atoms and an oxygen head group. Our DFT calculations demonstrate a 6.7 eV band gap in the stearic acid, with a negligible density of localized states in the band gap (Figure 1b).

To simplify the assembly of the graphene/molecule junction, we devised a device structure that only required one graphene/molecule interface. A graphene layer is covered by stearic acid, and then two contacts are placed on top of it (Figure 1c). In this geometry, carrier transport first occurs in the vertical direction between the source and graphene. Then, carriers move within the graphene in the lateral direction before crossing the molecular layer a second time to reach the drain. Due to the high barrier associated with transport within the molecular layer, the alternative lateral conduction pathway in the stearic acid film is significantly less efficient.

The fabrication of the proposed vc2Dmj structure requires the atomically precise assembly of the dielectric with molecular-scale thickness, as this parameter represents the critical channel dimension. We employed Langmuir–Blodgett deposition for this purpose, due to its proven ability to produce wafer-scale films with uniform single-layer thickness. In LB deposition, a sub-monolayer of molecules is compressed on the interface of an insoluble subphase [7]. Brewster-angle microscopy (BAM) was used to investigate the formation of stearic acid monolayers. A series of images were taken during compression (Figure 2a) that illustrate the transition from gaseous phases to liquid phases as the compression progresses. With further compression, the liquid phases transformed into a solid state. This evolution is supported by surface pressure measurements (Figure 2b) that indicate the compression until 30 mN/m, which agrees with previous reports for the solid phase of stearic acid [8].

The solid film of stearic acid resulting from this compression process was then transferred onto a graphene layer, which was positioned on Si/SiO_2_ substrates. For this purpose, a Langmuir–Blodgett process was utilized where the sample is moved out of the subphase while the lateral surface pressure is maintained (Figure 2c).

The morphology of the resulting graphene/stearic acid structure was examined using atomic force microscopy (AFM). A film-like structure is observed that exhibits regions of larger height. These protrusions represent over-compressed domains where stearic acid film corrugates in an out-of-plane direction [9]. Due to their larger separation from the graphene, these regions are not expected to contribute to the tunneling process.

A boundary was created in the stearic acid film, and the thickness of a single stearic acid layer on graphene was measured to be 1 nm, which agrees with previous results [10,11]. To demonstrate the robustness of the LB transfer process, we also deposited triple layers of stearic acid on graphene by conducting sequential LB steps. We observed the formation of a uniform layer with a thickness of 3 nm, as expected.

To establish the quality of the interface, we first investigated the heat transport in the graphene/molecular junction. Time-domain thermoreflectance (TDTR) shows a high thermal conductivity value, with a value of around 6000 W/mK (Figure 3a). The high out-of-plane thermal conductivity also confirmed by power-dependent Raman spectroscopy, which demonstrates the adherence to a linear relationship between Raman intensity and excitation power throughout the power range, without the indication of saturation as indicative for overheating [12]. The observed large thermal conductivity of the graphene/molecule junction is surprising, as stearic acid exhibits a low conductivity value [13]. We therefore hypothesize that graphene and stearic acid are forming a hybrid structure by bonding between the electronegative oxygen head group and the graphene basal plane.

To corroborate the formation of a stable hybrid, we immersed the junction into a liquid electrolyte and conducted electrochemical impedance spectroscopy. We observed a decreased heterogeneous charge transfer efficiency compared to bare graphene, thus confirming the presence of stearic acid acting as a barrier layer (Figure 3c). Even prolonged exposure to the electrolyte does not dissolve the stearic acid layer, corroborating its stability.

The hybrid graphene/molecular layer structure was further investigated by thermoelectric measurements; for this purpose, contacts were deposited on the graphene and the stearic acid. We extracted the Seebeck voltage by providing a temperature difference between the substrate and the top. A value of 2.99 µV/K was observed, which is significantly lower than expected for graphene at realistic carrier concentrations [14]. This observation corroborates the deviation of the graphene/molecule hybrid from pristine graphene. Finally, a ZT coefficient of 0.05 was extracted by Harmann-type measurements which is comparable to previous reports on the junctions between graphene and conductive molecules [15]. 

Our results suggest that stearic acid is modifying graphene’s electronic structure significantly. To investigate the effect of this modification on the quantum capacitance, we conducted DFT calculations. Compared to the density of states of pristine graphene (Figure 4a), we observed a large increase in DOS around the Fermi level (Figure 4b).

This increased quantum capacitance at an accessible voltage range imparts our vc2Dmj device with unique opportunities. To illustrate this capability, we measured the current–voltage characteristics of the device using source and drain contacts on the stearic acid. We observed a clear negative differential resistance (NDR) region in the devices (Figure 4c) [16]. Surprisingly, however, the device also demonstrates active negative resistance (ANR) behavior, where a positive current is flowing at negative applied voltages. Active negative resistance has not been observed in 2D material devices and points toward a novel operating mechanism. The observation that no NDR or ANR is observed when replacing the graphene layer with Au (Figure 4d) indicates the importance of the graphene component to the underlying mechanism.

Based on the presented observations, we propose the following operating mechanism: The application of a bias between source and drain will result in the accumulation of carriers in the graphene, due to the slow transport through the molecular dielectric. This accumulation will charge the graphene and increases its quantum capacitance. The change in quantum capacitance in turn increases the ability to store charges and enhances the injection current from the source (Figure 5a).

Temperature-dependent carrier transport measurements confirm the proposed transport process (Figure 5b). Our device exhibits a pronounced increase in device current with temperature, indicating thermionic emission. This operating mechanism is fundamentally different from the direct tunneling-based mechanism in conventional 2D NDR devices [17].

We can capture the vc2Dmj behavior through the effect of a changing capacitance on the measured current according to
$$i_{total}=i_{resistive}+i_{charging}=\frac{1}{R(V)}V+C(V)\frac{dV}{dt}$$
where R(V) is the voltage-dependent device resistance, C(V) is the changing capacitance, and dV/dt is the voltage scan rate.

Approximating the observed density of states of the graphene/molecule junction in Figure 4b with two Gaussian peaks is shown to properly reproduce the observed NDR and ADR regions (Figure 5c). The good agreement between the simple model and the experimental observation indicates the importance of variable quantum capacitance to the device performance.

The novel operating mechanism has several advantages over conventional NDR devices. First, thermionic emission into a smoothly varying DOS relaxes the requirements of device tolerances and materials quality compared to traditional resonant tunneling processes into specific states. This robustness in operation permits the realization of vc2Dmj devices at a large scale and from various material systems. Finally, the NDR is expected to be less sensitive to temperature, permitting operation at room temperature and above.

A second advantage of the device concept is the adjustability of the device properties through morphology changes. Capacitance variation, as the operating mechanism, is sensitive to the total electrostatic capacitance of the device that contains contributions from electrodes and surfaces. Therefore, modifying the device geometry is expected to control the electrostatics of its operation. We demonstrate this ability by increasing the contribution of the quantum capacitance in the vc2Dmj and enhancing the NDR and ADR.

The total capacitance of the vc2Dmj can be considered as follows:$$\frac{1}{C_{total}}=\frac{1}{C_{quantum}(V)}+\frac{1}{C_{es}}$$

Minimization of the electrostatic capacitance will increase the effect of voltage change on the total capacitance. This decrease in the electrostatic capacitance can be achieved by increasing the spacing between source and drain contacts to the graphene.

We utilized single-layer molecular dielectrics and triple-layer dielectrics to increase the spacing and observe the same non-monotonic behavior for both structures, confirming the robustness of the process (Figure 5d). As expected, the vc2Dmj with larger vertical separation exhibits a better ADR and NDR performance, and the NDR reaches a peak-to-valley current ratio (PVCR) of 2. This parameter is comparable to previously reported graphene-based resonant tunneling devices and exceeds the performance of many 2D heterojunctions [18]. In the future, this parameter could be further enhanced through electrostatic control through a gate terminal.

## 4. Conclusions

We demonstrated the realization of a novel electronic device design that leverages the non-monotonic quantum capacitance of a graphene/molecular layer junction. Scalable production and a unique active negative resistance region make this vc2Dmj diode promising for non-traditional circuit designs in future electronics.

## Figures and Tables

**Figure 1 nanomaterials-14-00972-f001:**
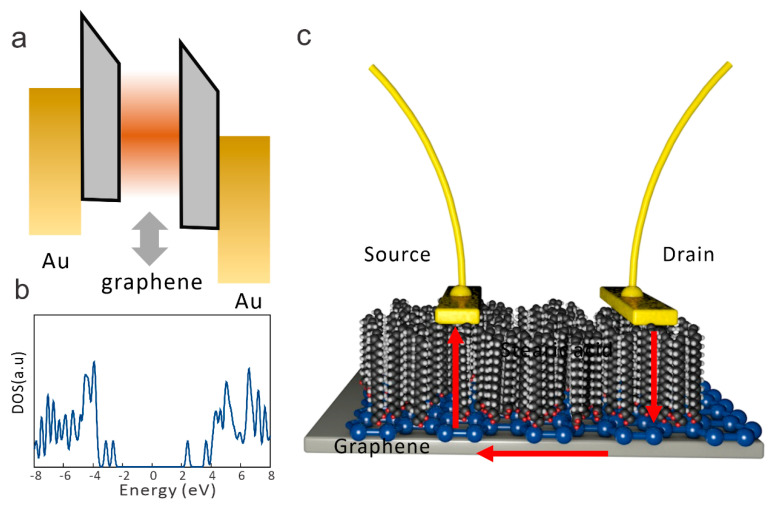
Concept of vc2Dmj device structure: (**a**) band structure of proposed device showing variable capacitance graphene layer between Au source and drain contacts, (**b**) DFT calculated density of states for stearic acid (inset) depiction of stearic acid, and (**c**) schematic of device realization with indication of current pathway.

**Figure 2 nanomaterials-14-00972-f002:**
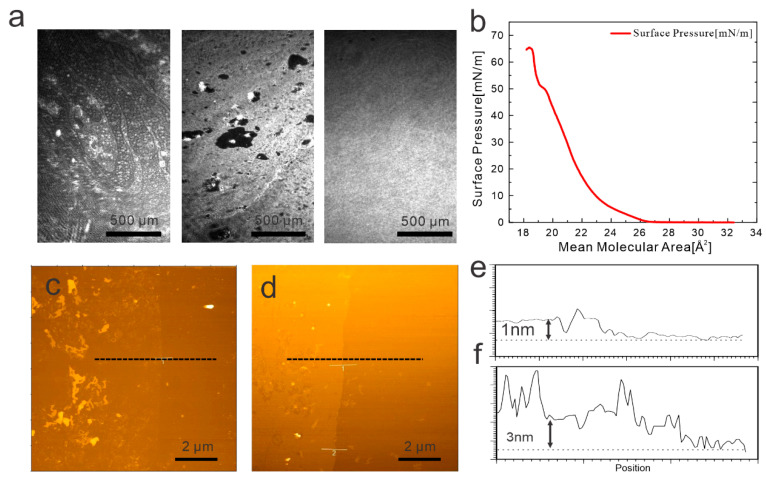
Characterization of dielectric molecule deposition: (**a**) series of Brewster-angle micrographs at different compression values showing condensation of molecular layer, (**b**) corresponding surface pressure curve vs. compression, (**c**) atomic force micrograph of deposited single molecular layer, (**d**) atomic force micrograph of deposited triple molecular layer, (**e**) cross-section corresponding to (**c**), and (**f**) cross-section corresponding to (**f**).

**Figure 3 nanomaterials-14-00972-f003:**
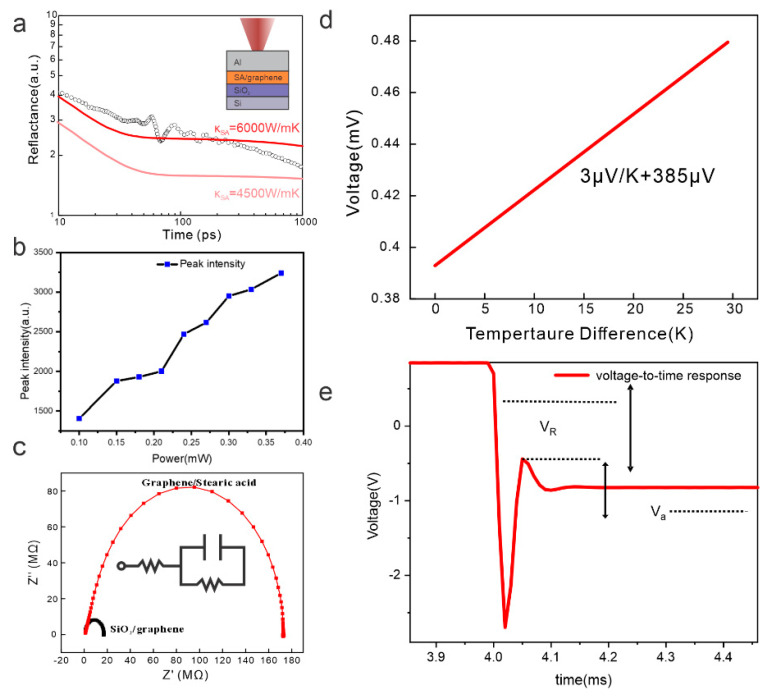
Characterization of graphene/stearic acid junction: (**a**) time-domain thermoreflectance (TDTR) measurements with least-square fitting to two 4-layer models with differing thermal conductivity (more details in the Appendix A); (**b**) Raman intensity vs. power plot; (**c**) electrochemical impedance spectroscopy plot for graphene layer and junction with fit to RC circuit in (inset); (**d**) Seebeck voltage measurement; and (**e**) Harmann measurement of ZT.

**Figure 4 nanomaterials-14-00972-f004:**
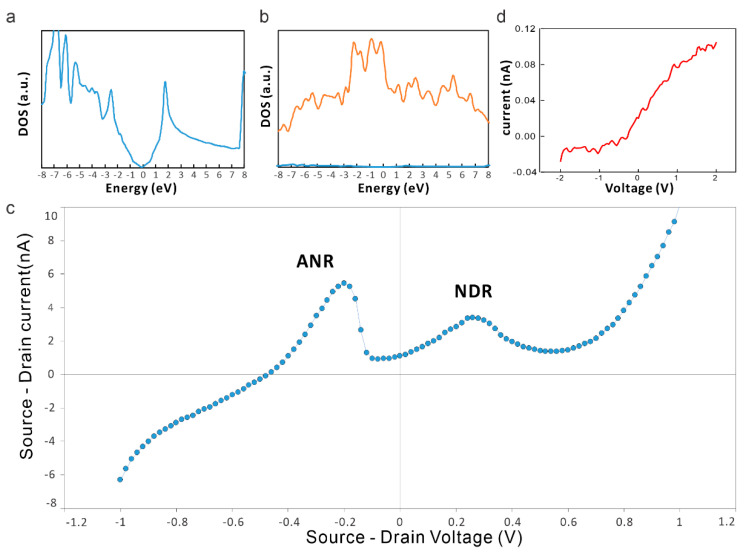
Carrier transport in vc2Dmj: (**a**) DFT calculated density of states of bare graphene; (**b**) density of states for graphene/stearic acid junction; (**c**) representative current–voltage characteristics showing negative differential resistance (NDR) and active negative resistance (ANR); and (**d**) current–voltage characteristics of vc2Dmj, where graphene is replaced by a Au film.

**Figure 5 nanomaterials-14-00972-f005:**
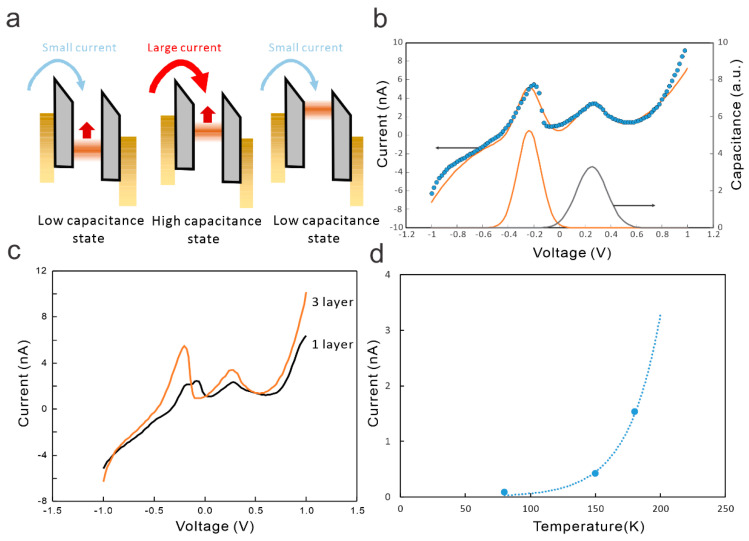
Mechanism of vc2Dmj operation: (**a**) schematic of current flow under different charging conditions of graphene; (**b**) fitting of current–voltage characteristics to simple quantum capacitance model; (**c**) comparison of carrier transport for junction consisting of graphene and single- or triple-layer stearic acid, respectively; and (**d**) temperature dependent current at 0.5 V with fit to thermally activated emission.

## Data Availability

The data that support the findings of this study are available from the corresponding author upon reasonable request.

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
