# Peer review of "Harnessing Quantum Capacitance in 2D Material/Molecular Layer Junctions for Novel Electronic Device Functionality"

_nanomaterials, 2024, doi:10.3390/nano14110972_

Round 1

Reviewer 1 Report

Comments and Suggestions for Authors

Subject: Review of Manuscript Submission Nanomaterials-3014215

I have carefully reviewed the manuscript titled "Harnessing Quantum Capacitance in 2D Material/Molecular Layer Junctions for Novel Electronic Device Functionality” submitted to Nanomaterials for consideration. This study makes a variable capacitance 2D molecular junction (vc2Dmj) diode through the scalable integration of graphene and single layers of stearic acid, exhibiting NDR with substantial peak-to-valley ratio even at room temperature and an active negative resistance region. It can be accepted after major revision.

1.       The dielectric molecule deposition looks dense but not uniform, as shown in figure 2, including AFM, and the corresponding topographic mapping. Will this unevenness affect the effectiveness of the device? It is recommended that the authors discuss it.

2.       The fitted curve for TDTR in figure 3a is not ideal. The authors should refit it. It is recommended to re-fit it using a two-parameter or three-parameter and label the details of the fitted parameters in the figure.

3.       Figure 3c should be modified. i The Y-axis label should be completed. ii The inserted fitted circuit diagram should be labeled with the labels of the different components. In figure 3d, the complete fitted curve should be listed,  not only the slope.

4.       In figure 4, 4a, and 4b should be assembled into a single diagram.  Individual listings do not convey much significance.

5.       The authors should emphasize the pics' qualities and clarities, which is essential to meet the requirements of the journal. It seems should be 300 dpi at least.

Reviewer 2 Report

Comments and Suggestions for Authors

Impressive work - only have a few questions for clarification that may be considered for addition into the work:

1) Were any experiments performed or considered for different metal contacts? 

2) What were the environmental conditions (light, humidity, heat) during the observation of ANR in the graphene:stearic acid device?

Round 2

Reviewer 1 Report

Comments and Suggestions for Authors

It can be accepted in its present form.